# Multifunctional Gelatin-Nanoparticle-Modified Chip for Enhanced Capture and Non-Destructive Release of Circulating Tumor Cells

**DOI:** 10.3390/mi13030395

**Published:** 2022-02-28

**Authors:** Linying Xu, Tiantian Ma, Kelin Zhang, Qilin Zhang, Mingxia Yu, Xingzhong Zhao

**Affiliations:** 1Key Laboratory of Artificial Micro- and Nano-Structures of Ministry of Education, School of Physics and Technology, Wuhan University, Wuhan 430072, China; xulinying@whu.edu.cn (L.X.); 9176605@163.com (K.Z.); qilin-zhang@whu.edu.com (Q.Z.); 2Department of Clinical Laboratory, Zhongnan Hospital of Wuhan University, Wuhan 430071, China; 2020203030021@whu.edu.com

**Keywords:** gelatin nanoparticles, circulating tumor cells, cell capture, non-destructive release

## Abstract

Circulating tumor cells (CTCs) in cancer patients’ peripheral blood have been demonstrated to be a significant biomarker for metastasis detection, disease prognosis, and therapy response. Due to their extremely low concentrations, efficient enrichment and non-destructive release are needed. Herein, an FTO chip modified with multifunctional gelatin nanoparticles (GNPs) was designed for the specific capture and non-destructive release of CTCs. These nanoparticles share a similar dimension with the microvilli and pseudopodium of the cellular surface; thus, they can enhance adhesion to CTCs, and then GNPs can be degraded by the enzyme matrix metalloproteinase (MMP-9), gently releasing the captured cells. In addition, the transparency of the chip makes it possible for fluorescence immunoassay identification in situ under a microscope. Our chip attained a high capture efficiency of 89.27%, a release efficiency of 91.98%, and an excellent cellular viability of 96.91% when the concentration of MMP-9 was 0.2 mg/mL. Moreover, we successfully identified CTCs from cancer patients’ blood samples. This simple-to-operate, low-cost chip exhibits great potential for clinical application.

## 1. Introduction

In recent decades, cancer diagnostic modalities have been mostly based on serum screening and tissue biopsy. However, the biomarkers in serum are not specific enough, [1] and tissue biopsy may be harmful. A more accurate and non-invasive cancer diagnosis is attracting great attention. For example, liquid biopsy, which is quick and non-invasive, could be a potential alternative detection method for early cancer diagnosis. The biomarkers in peripheral blood, such as circulating tumor cells (CTCs), extracellular vesicles (EVs), cell-free nucleic acids (cfNAs), and tumor-educated platelets (TEPs), could be detected as tumor-associated components [2]. In regard to these biomarkers, it is difficult to separate EVs from serum or plasma, and the protein in the background may be interferential [3]. cfNAs include both cfDNA and cfRNA, while ctDNA (relating to tumor cell DNA) accounts for only 0.01% of all cfDNA [4]. This extremely low concentration brings a great challenge to separating ctDNA from non-tumor DNA. Moreover, the diagnostic potential of TEPs is usually evaluated by mRNA sequencing [5], but the complicated protocol [6] of the detection method and the high cost of testing lead to difficulty in transferring this method to clinical applications.

Compared to the aforementioned biomarkers, CTCs, deriving from a primary tumor mass and entering the blood circulation, which happens at the early stage of malignant progression [7,8,9,10], may contain genetic information about the primary tumor [11]. Recently, increasingly more cost-effective and relatively simple methods have been proposed to distinguish them from background normal blood cells and to isolate them for further analysis. In a study by Nagrath et al. [12], CTCs were detected in 115 of 116 cancer patients’ peripheral blood samples, and they are probably a potential alternative to invasive biopsies. Additionally, they can be repeatedly detected to monitor therapeutic efficacy and predict prognosis [13]. Unfortunately, the extremely low levels of CTCs in peripheral blood (one tumor cell in 10^6^–10^7^ normal blood cells) brings a great challenge in the enrichment of CTCs [14].

Significant research efforts have been devoted to enrich and isolate the CTCs in cancer patients’ blood. The traditional physical methods based on cellular size, electric properties, density, and deformability may cause cell loss and low purity. In order to solve these problems, researchers have developed diverse mechanisms of CTC enrichment, such as immunomagnetic separation [15,16], microfluidic chips [17,18,19,20], and micro-filter devices [21,22]. Although these methods have their own advantages, the complicated experimental operations and the long analytical time limit their further application. In recent years, many researchers have been devoted to nanomaterials and nanostructured substrates for CTC enrichment. A high ratio surface area will provide more bonding sites for capture agents, and nanostructured substrates will increase the affinity with CTCs, as it has been demonstrated that the capture efficiency of CTCs could be enhanced [23]. Dong, J. et al. [24] mentioned many kinds of nanoparticles in their review, such as Fe_3_O_4_ nanoparticles with a diameter of 25 nm, TiO_2_ nanoparticles with a diameter of 400 nm, and candle soot with a diameter of 19–43 nm. These nanostructures can enhance the performance of the CTC capture yield (>80%) due to their similar dimensions with cellular surface components (such as filopodia) and extracellular matrix scaffolds [25]. Unfortunately, these nanoparticle-based substrates have poor performance in viable cell release and further downstream analysis.

Herein, we designed a 1 cm × 1 cm transparent nanostructured substrate with a gelatin nanoparticle (GNP) coating on FTO glass for cancer cell capture and non-destructive release. A schematic diagram of the GNP-coated chip is shown in Figure 1.

The GNP chip has several merits: First, gelatin material is biocompatible and non-toxic; it is a mixture of proteins and peptides extracted from animal collagen; and due to the ability to induce cell adhesion and proliferation, gelatin is widely used as a coating material on substrate for adherent cell capture [26,27]. The synthesis of gelatin nanoparticles and the chemical modification of antibodies are simple in comparison to the complex chemical etching methods for substrate fabrication, such as soft lithography [21] and dry etching [28]. Moreover, the centralized size distribution of GNPs is between 150 and 300 nm (Figure 2c), which can enhance the local topography and provide a large surface area to achieve better capture efficiency. Except for these advantages, GNPs can be easily degraded by a low-concentration enzyme matrix metalloproteinase (MMP-9) solution [29], mildly cutting off the bond between the substrate and cells with negligible damage to the cell; thus, the released cells have high viability for subsequent analysis. Compared to other techniques, it is worth noting that our biocompatible microchip is low cost, transparent, and visible for further immunofluorescence identification.

## 2. Materials and Methods

### 2.1. Materials and Instruments

Sodium hydroxide (NaOH) was purchased from Sinopharm Chemical Reagent Co., Ltd., Shanghai, China. Gelatin type B, glutaraldehyde solution (Grade I, 50%), acetone, matrix metalloproteinase-9 (MMP-9), 2-(4-morpholino)-ethane sulfonic acid (MES), 1-ethyl-3-[3-(dimethylamino) propyl] carbodiimide hydrochloride (EDC), 3-Aminopropyltriethoxysilane (APTES), paraformaldehyde (PFA, 36% in water), Triton X-100, 4,6-Diamidino-2-phenylindole dihydrochloride (DAPI), fluorescein diacetate (FDA), N-hydroxysuccinimide (NHS), propidium iodide (PI), bovine serum albumin (BSA), and Tween 20 were purchased from Sigma-Aldrich. Phosphate-buffered saline (PBS), Dulbecco’s modified Eagle’s medium (DMEM, Hyclone, high glucose), and lymphocyte separation medium (LSM) were obtained from Wuhan Era Bochi Science & Technology Co., Ltd., Wuhan, China. Streptavidin (SA) was purchased from Thermo Fisher. Biotinylated anti-human EpCAM was obtained from R&D System. Fluorescein isothiocyanate (FITC)-labeled streptavidin, phycoerythrin (PE)-labeled anti-cytokeratin (CK), and fluorescein isothiocyanate (FITC)-labeled anti-human CD45 were obtained from BD Biosciences.

The DLS results of GNPs were characterized by Zetasizer (Zetasizer Nano ZSP), TEM images were obtained using JEM-2010 FEF transmission electron microscope, and SEM images were produced by Hitachi S4800 FEG electron microscope.

### 2.2. Cell Culture and Counting

We used breast cancer cell line MCF-7, colorectal cancer cell line HCT116, and white blood cells (WBCs) to optimize the experimental conditions. Cancer cell lines were cultured in DMEM medium with 10% fetal bovine serum (FBS) and 1% penicillin and streptomycin at humidified atmosphere of 37 °C and 5% CO_2_. Hemocytometer was used to calculate the cell concentration.

### 2.3. Gelatin Nanoparticle Synthesis, Functionalization, and Characterization

GNPs were synthesized on the basis of a two-step desolvation method that we have previously reported on [30]. At first, 0.625 g gelatin type B powders were dissolved in 12.5 mL DI water at 50 °C. Then, 12.5 mL acetone was added into the solution at 6.25 mL/min and stirred at 600 rpm during the heating. The stirring was turned off when the addition of acetone finished. After one minute, the gelatin nanoparticles at the bottom became orange, and the supernatant was completely removed. Hydrogel sediment was then dissolved at 50 °C with 12.5 mL DI water again. The pH of the solution was adjusted to 10 by adding sodium hydroxide dropwise. After that, acetone was added into the solution at 1 mL/min, with a temperature of 50 °C and stirring at 1000 rpm, until the solution appeared cloudy, caused by the scattering of synthesized GNPs. Subsequently, another 1 mL acetone solution containing 2.5% glutaraldehyde was added to crosslink the GNPs for 16 h at room temperature while still stirring at 1000 rpm. Finally, the obtained GNPs were collected by centrifugation and completely washed with DI water, and then they were suspended in DI water and stored at 4 °C.

Then, the GNPs were sufficiently washed with MES solution (0.195 g MES and 0.293 g NaCl in 10 mL DI water) and treated with EDC (4 mg)/NHS (6 mg) dissolved in 1 mL MES for 30 min incubation to functionalize GNPs.

Synthesized GNPs were characterized by dynamic light scattering (DLS, Nano-Zen 3600, Malvern Instruments, Malvern, UK) and scanning electron microscope (SEM, S4800, Hitach, Hiroshima, Japan).

### 2.4. Chemical Modification of the GNP-Coated Chip

First, the chip was treated with 5% APTMS in anhydrous ethanol for 1 h to functionalize the surface with alkoxysilane molecules. After being washed three times with ethanol, the microchip was treated with functionalized GNP solution for 1 h so that the gelatin nanoparticles absorbed onto the chip surface through peptide linkage.

### 2.5. Conjugation of SA and Antibody onto the Substrate

The substrate was washed with PBS and treated with 50 µg/mL streptavidin at 4 °C overnight. After being washed with PBS to remove the redundant SA, the chip was then incubated with biotinylated anti-EpCAM (5 µg/mL) at room temperature for 1.5 h in preparation for the cell capture experiment.

### 2.6. Cell Capture and Release

To mimic the real experiment, we used MCF-7 and HCT116 cell lines (WBCs as a negative control) spiked in PBS and processed blood samples. First, in the process of cell subculture, the digested cells were collected and spiked in PBS or mononuclear cell suspension. Subsequently, the cells were introduced to the substrate surface for incubation. After being washed with PBS, we used a microscope to observe the captured cells that remained on the substrate and recorded the number. As Equation (1) shows, the ratio of the captured cells to spiked cells was taken as capture efficiency.

MMP-9 solution was chosen to degrade the GNPs, releasing the captured cells from the chip. The viability of released cells was assessed by PI and FDA, and the live cells were stained by FDA and dead cells by PI. Release efficiency and cell viability are defined according to Equations (2) and (3). In addition to investigating the release performance of cancer cell lines, we also explored the release performance of monocytes in peripheral blood to evaluate capability of CTC release and its viability test.
(1)$$\mathrm{capture}\;\mathrm{efficiency}=\frac{\mathrm{captured}\;\mathrm{cells}}{\mathrm{spiked}\;\mathrm{cells}}$$
(2)$$\mathrm{release}\;\mathrm{efficiency}=\frac{\mathrm{captured}\;\mathrm{cells}-\mathrm{remained}\;\mathrm{cells}}{\mathrm{captured}\;\mathrm{cells}}$$
(3)$$\mathrm{cell}\;\mathrm{viability}=\frac{\mathrm{alive}\;\mathrm{cells}}{\mathrm{alive}\;\mathrm{cells}+\mathrm{dead}\;\mathrm{cells}}$$

### 2.7. CTC Capture from Clinical Samples and Staining

First, we needed to pretreat the peripheral blood samples to enrich mononuclear leucocytes containing CTCs and dispel erythrocytes. Briefly, the blood sample was well diluted with PBS in a ratio of 1:1 and then carefully added onto the surface of lymphocyte separation medium (LSM) in a 15 mL centrifuge tube. After centrifugation at 400× *g* for 30 min, due to the different densities, there were 4 layers in the tube: serum, mononuclear leucocytes, LSM, and erythrocytes. The upper and bottom layers were discarded. Then, we transferred about 1 mL of the mononuclear leucocytes into another tube and then added PBS to achieve a 2 mL solution. After centrifugation at 310× *g* for 10 min, we suspended the cells in the bottom into 1 mL PBS for capture.

The obtained cell suspension was added to the chip dropwise for 1.5 h incubation. Then, the chip was gently washed with PBS; treated with 4% PFA for 10 min to fix the cells, 0.3% Triton-X100 for 10 min to enhance membrane permeability, and 3% BSA in PBS for 30 min for blocking; and then stained with PE-labeled anti-CK overnight at 4 °C and DAPI for 10 min at RT. After that, CTCs were distinguished from WBCs according to immunofluorescence of DAPI, anti-CK (PE), and anti-CD45 (FITC). DAPI is for nuclear staining, and anti-CK is a specific marker for CTCs, the anti-CD45 for WBCs. The cells showing DAPI(+)/CD45(+)/CK(−) are recognized as WBCs, while DAPI(+)/CD45(−)/CK(+) as CTCs.

## 3. Results and Discussion

### 3.1. Characterization and Cell Capture Performance of GNP-Modified Chip/Bare FTO Chip

In this study, a GNP-modified chip was fabricated. The morphologies of the GNP-coated substrate and the bare FTO substrate were characterized by SEM, and they are displayed in Figure 2a,b. On the substrate, the GNPs were well distributed, with a centralized size between 150 and 300 nm (Figure 2c). The GNPs enhanced the local topography, and the AFM figures are shown in Appendix A. Thus, compared with the bare FTO chip, the GNP-modified chip may achieve better capture efficiency. The zeta potential of the GNPs was –33.67, confirming the stability of the synthesized GNPs (Figure 2d).

To validate the cell capture performance of the GNP-modified chip, MCF-7 cell lines were chosen to mimic CTCs in maternal blood. After the cells were sufficiently distributed in the PE tube and the concentration of the cell suspension was calculated, about 1000 cells were spiked onto the GNP-coated chip, incubated for 1.5 h, and gently washed with PBS, and the remaining cells on the chip were counted. The number of remaining cells divided by the number of spiked cells was defined as cell capture efficiency to assess cell capture performance, and the bare FTO chip was used as the control. As Figure 3a-i,b-i show, compared with the bare FTO chip, more cells were captured on the GNP-modified chip, suggesting that the latter obtained higher cell capture efficiency. SEM photographs were used (Figure 3a-ii,b-ii) to further study the interaction between the GNPs and the cells. The target cell on the GNP-coated chip stretched extensively and was tightly attached to the GNP substrate, while the cell on the bare FTO had less pseudopodia, demonstrating that the enhanced local topography caused by the GNPs may be an essential factor for the affinity of the cell binding to the substrate.

### 3.2. Optimization for Cell Capture Performance

To obtain the optimal conditions for cell capture performance, five concentrations (0, 0.5, 1.0, 2.5, and 5.0 mg/mL) of the GNP-modified chip were prepared to validate the effect. The experimental protocols were carried out as previously mentioned. Appendix A presents the characterization and cell capture performance of the five different concentrations of GNP-modified chips, as well as specific capture efficiency (Appendix A). As Figure 4a displays, cell capture efficiency increased as the concentration of the GNPs increased until 2.5 mg/mL; a higher concentration makes no contribution to capture yield improvement, so the optimal concentration of GNPs for cell capture was 2.5 mg/mL.

Then, we investigated different incubation times (15, 30, 60, 90, and 120 min) for cell capture. The correlation between the incubation time and the cell capture efficiency is shown in Figure 4b. The maximal ratio of efficiency was reached when the incubation time was 90 min.

Furthermore, to assess the effect of antibody specificity on capture efficiency, two EpCAM-positive cancer cell lines (MCF-7 and HCT116) were chosen to imitate circulating tumor cells, and WBCs were used as a negative control to evaluate nonspecific interactions. Detailed information, such as the number of spiked cells and the number of captured cells, is shown in Appendix A. However, there are mesenchymal CTCs with no EpCAM, and they are of importance in metastasis. By combining mesenchymal markers (such as anti-PTK7) with anti-EpCAM, our platform could also be used to capture CTCs of both epithelial and mesenchymal phenotypes.

According to Figure 4c, MCF-7 and HCT116 cells achieved a high capture efficiency of 89.57% and 88.17% respectively, and the nonspecific attachment efficiency was 5.75%. This suggests that anti-EpCAM is sensitive and specific for EpCAM-positive CTCs.

Using the optimal experimental parameters, MCF-7 cells were spiked into PBS and processed healthy human blood successively to mimic real CTC capture performance. First, cells were stained with FDA so that they would be different from WBCs (Appendix A). Subsequently, a number of stained cells (Appendix A) were spiked into PBS and processed blood separately. After incubation for 90 min, we counted the remaining cells on the chip to investigate its performance. Figure 4d,e show that a high capture efficiency could be reached in both PBS and processed blood.

### 3.3. Cell Release and Viability Assay

The GNPs can take effect both in the process of cell capture and release. During the cell release procedure, GNPs are sensitive to matrix metalloproteinase-9 (MMP-9) and can be digested at a low concentration of MMP-9; thus, the released cells have a high viability for downstream analysis. Different concentrations of MMP-9 solution (0 mg/mL, 0.01 mg/mL, 0.1 mg/mL, 0.2 mg/mL, and 0.5 mg/mL) were explored to improve cell line release performance. After treatment with MMP-9 solution for 30 min at 37 °C, the cells that remained on the substrate were counted, and the release efficiency could be obtained by Equation (2). The released cells were centrifuged and then labeled with fluorescein diacetate (FDA) and propidium iodide (PI). With the microscope, we calculated the cell viability according to Equation (3).

Figure 5a shows that, as the concentration of the MMP-9 solution increased, the efficiency of cell release rose and then tended to be flat after the turning point of 0.2 mg/mL. Cells released from the substrate by different concentrations of MMP-9 solution displayed high viability (Figure 5b and Appendix A), demonstrating that a low concentration of MMP-9 solution is mild and non-toxic to cells. Taking both cell viability and release efficiency into consideration, the optimal concentration of MMP-9 was 0.2 mg/mL, with a release efficiency of 91.98% and an excellent cellular viability of 96.91%.

Figure 5c shows the detail that, with an increase in MMP-9 (0.2 mg/mL) treatment time, the fluorescence intensity gradually weakens. At 30 min, the intensity almost disappeared, indicating that the gelatin nanoparticles were completely degraded at this time.

We also explored the released performance of peripheral blood mononuclear cells (PBMCs) from a gastric cancer patient, and the result is shown in Appendix A and Appendix A. The viability of the PBMCs from the gastric cancer patient was 78.43%, suggesting that CTCs released through our platform also have high activity.

In a subsequent experiment, we carried out a cell viability assay using released cells to investigate proliferative performance. The cells captured by the GNP-modified chip were released using 0.2 mg/mL MMP-9 solution, and then, through centrifugation, the cells were collected and re-cultured with DMEM. At the times of 0 h, 24 h, 36 h, and 48 h, we tested cell viability by FDA/PI staining. Figure 5d shows the good proliferative performance of the released cells, which sufficiently confirms the non-destructive and biocompatible properties of our platform.

### 3.4. CTC Capture and Identification from Cancer Patients’ Blood

After a series of optimizations, we finally applied our GNP-coated chip to isolate CTCs from cancer patients’ blood samples. Three blood samples from breast cancer patients, four blood samples from colorectal cancer patients, and two blood samples from healthy donors were used to validate the feasibility of CTC capture during intermediate stage cancer (detailed information about the patients and healthy individuals is shown in Appendix A). As Figure 6a shows, the cells present DAPI (+)/CD45(+)/CK(−) are recognized as WBCs, while DAPI (+)/CD45(−)/CK(+) are CTCs. According to the experiment’s results, the number of CTCs identified from 1 mL of blood sample ranged from 3 to 13, while no CTCs were identified in the healthy donors’ blood samples (Figure 6b).

## 4. Conclusions

In summary, we developed a GNP-modified chip, which is biocompatible and transparent, for CTC capture, non-destructive release, and identification. First, multifunctional GNPs were synthesized and then drafted onto the FTO glass chip to increase surface roughness and enhance the attachment between the cells and substrate. In our experiments, capture efficiency reached up to 89.27%. Furthermore, GNPs were mildly degraded by 0.2 mg/mL of MMP-9 solution, with CTCs released from the substrate at a high efficiency of 91.98% and a viability of 96.91%. Last but not least, our device was successfully applied to capture CTCs from advanced cancer patients’ peripheral blood samples. Seven of these were identified with CTCs, while two healthy blood samples were not, proving that our device may have potential application value in CTC detection and related clinical analysis. Thus, this work contributes to cancer diagnosis and therapeutic monitoring.

## Figures and Tables

**Figure 1 micromachines-13-00395-f001:**
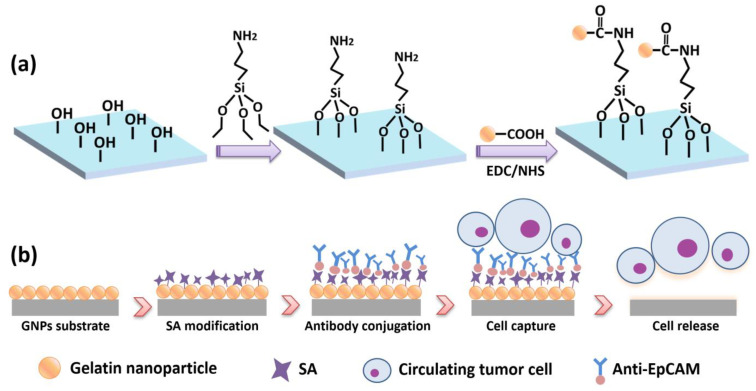
Schematic diagram of GNP-coated chip. (**a**) GNP substrate fabrication process; (**b**) antibody modification of the substrate for CTC capture and release.

**Figure 2 micromachines-13-00395-f002:**
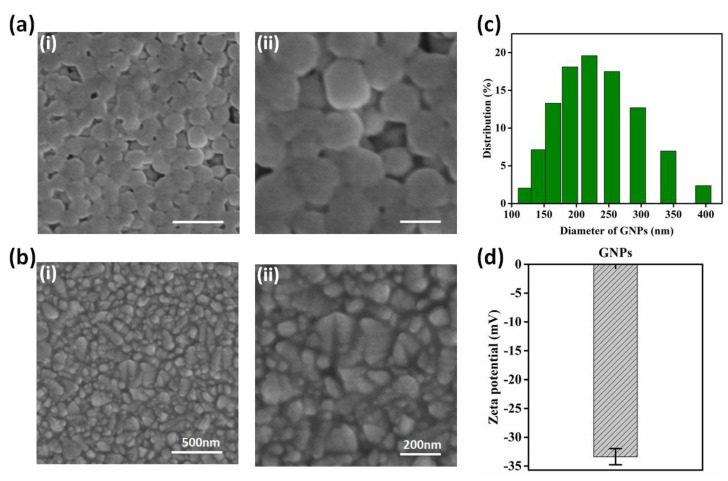
Characterization of GNPs and substrate. (**a**,**b**) SEM images of GNPs–coated substrate and bare FTO substrate; scale bars in (**i**) and (**ii**) are 500 nm and 200 nm, respectively. (**c**) DLS analysis of GNP diameter distribution; the centralized size is between 150 and 300 nm. (**d**) Zeta potential of GNPs, showing its stability (*n* = 3).

**Figure 3 micromachines-13-00395-f003:**
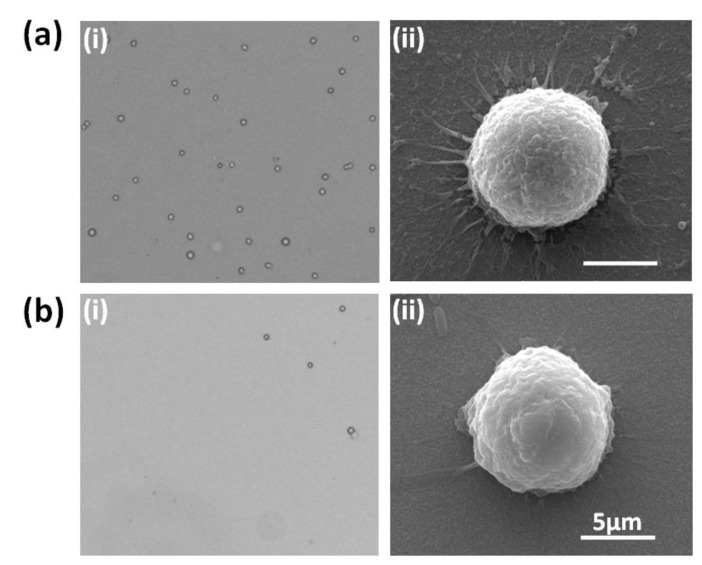
Cell capture performance of GNP-modified chip and bare FTO chip. (**a**-**i**) Microscope image of cell capture performance by GNP-coated chip; (**a**-**ii**) SEM image of single cell attached on GNP-coated chip; (**b**-**i**) microscope image of cell capture performance by bare FTO chip; (**b**-**ii**) SEM image of single cell attached on bare FTO chip.

**Figure 4 micromachines-13-00395-f004:**
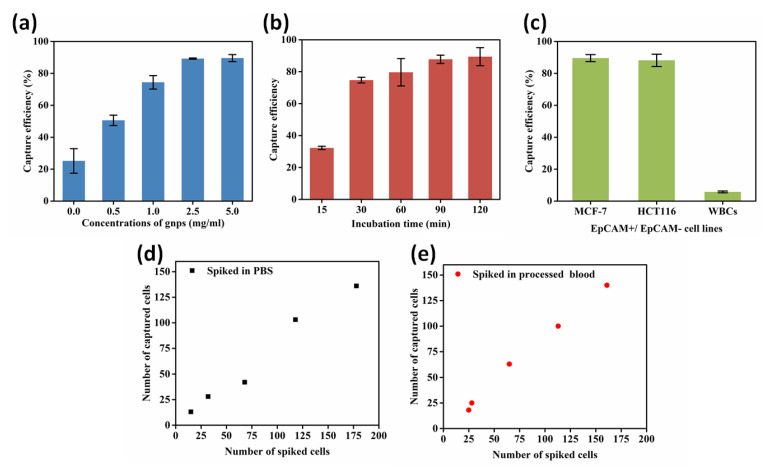
Optimization of different experimental conditions to enhance cell capture performance. (**a**) The effect of 5 different concentrations of GNPs on capture efficiency for MCF-7 cells. (**b**) Evaluation for MCF-7 cell capture efficiency with different incubation times (30, 60, 90, and 120 min). (**c**) Capture efficiency of MCF-7 and HCT116, and WBCs response to anti-EpCAM grafting chip. (**d**) A number of MCF-7 cells spiked in PBS to validate capture efficiency. (**e**) A number of MCF-7 cells spiked in mononuclear cells to validate capture efficiency.

**Figure 5 micromachines-13-00395-f005:**
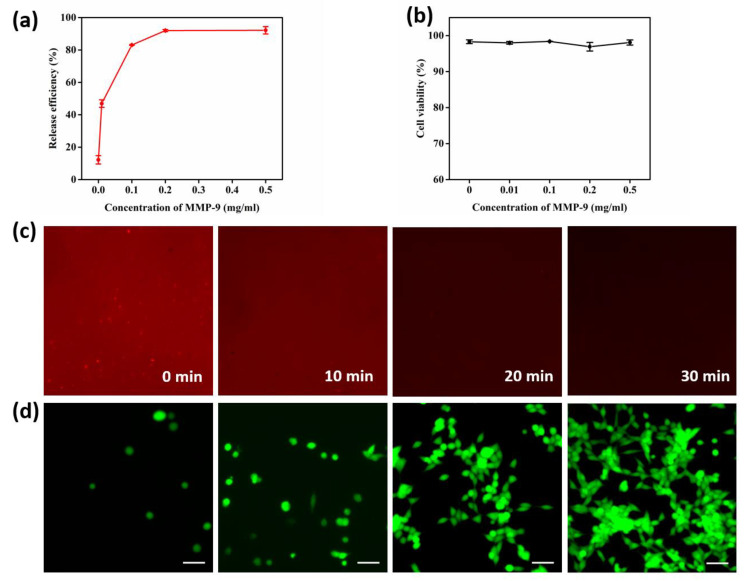
Optimization of MMP-9 concentration and cell release performance. (**a**) Cell release efficiency treated with different concentrations of MMP-9 (0, 0.01, 0.1, 0.2 0.5 mg/mL). (**b**) Cell viability after treatment with different concentrations of MMP-9 for 30 min at 37 °C. (**c**) Fluorescence images of PE-SA-conjugated GNP substrate treated with MMP-9 for 0, 10, 20, and 30 min. (**d**) The viability of released cells using 0.2 mg/mL of MMP-9 solution and re-cultured at times of 0 h, 24 h, 36 h, and 48 h.

**Figure 6 micromachines-13-00395-f006:**
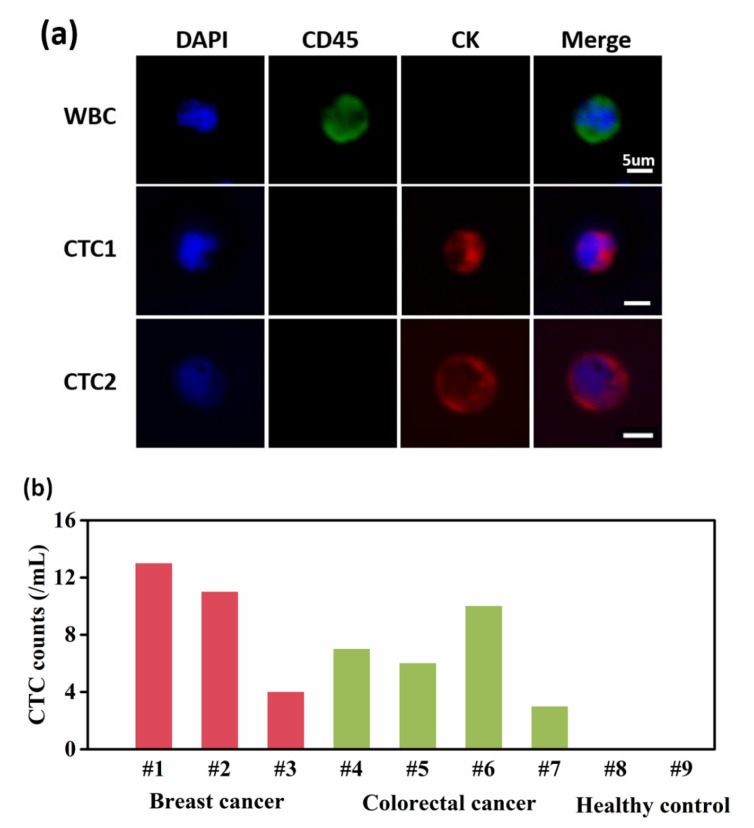
Identification of CTCs from 7 advanced cancer patients’ blood samples and 2 healthy donors’ blood samples. (**a**) Three color immunofluorescent micrographs (DAPI, FITC l-anti-CD45, PE-anti-CK) of WBCs and CTCs. (**b**) Number of CTCs from cancer patients’ blood samples and healthy donors’ blood samples.

## Data Availability

Not applicable.

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
