# Peer review of "Multifunctional Gelatin-Nanoparticle-Modified Chip for Enhanced Capture and Non-Destructive Release of Circulating Tumor Cells"

_micromachines, 2022, doi:10.3390/mi13030395_

Round 1

Reviewer 1 Report

This paper reported a platform for capturing and releasing CTCs effectively. The authors used gelatin nanoparticle-coated chip to improve capturing efficiency and provide the way to release cells nondestructively from the chip. However, there are several defects which are needed to be improved before consideration for publication. The comments for the manuscript are listed below:

  1. While the title says non-destructive release of CTCs, no result showed the release of CTCs from the chip. Only stable cell lines were demonstrated. For claiming the capability of CTC release and further study with it, the authors have to show the results of CTC release and its viability test without the fixation because CTCs must be hard to culture in vitro even after long-term, stressful processing. Also, the cell viability experiments with cell lines have no systematic analyses to show how many cells were viable after capturing and releasing. Please show the systematic analytical data for this.
  2. Currently, a lot of publications have mentioned that there are mesenchymal CTCs having no EpCAM and they are of importance in metastasis. In this paper, EpCAM was used as one and only receptor for capturing CTCs. Please give us the strategy to overcome this.
  3. A variety of platforms for CTC counting and immunostaining with a few markers have been developed so far. What are the most distinguishable points of this platform?
  4. There are a lot of grammar errors throughout the manuscript. Page1 Line30 “Such as liquid biopsy” to “Liquid biopsy”, Page1 Line 43 “enter” to “entering”, Page2 Line50 “levers” to “levels”, and others. Please revise the whole manuscript thoroughly. Also, please give the full name and abbreviations of the words when they first appeared such as MMP-9 and FDA.

Reviewer 2 Report

This manuscript can be published after a large revision.

In this manuscript, the authors developed a gelatin nanoparticles modified chip for circulating tumor cells capture and release. It is a very simple method and the capture efficiency is satisfied. After capture, the GNPs can be digested by MMP9 to gently release the CTCs with high cell viability. At last, the chip was applied to the clinical samples and demonstrated the great potential for clinic application. However, there are some questions about this manuscript.

There are 32 references, but 11 of the 32 references are published by authors themselves. Thousands of typical papers about the CTCs have been published and many of them are very typical literature. But I never read any paper like this manuscript. It is doubted that the authors are deliberate to increase their paper citations. Please give a reasonable explanation about this behavior.

  1. Page 2, Line 64, what is the “Jiantong D et al”? Please check if it is correct.
  2. Page 4, Line 134, there are many –COOH and -NH2 on the gelatin, when the authors treated the GNPs with EDC and NHS, will the GNPs react with other GNPs and aggregate as large particles?
  3. Page 4, Line 153. What is the number of MCF-7, HCT116, and WBCs used in the assay? Please provide the essential details.
  4. Page 4, Line 167. Why do the authors need to pretreat the peripheral blood? Can you directly process the whole blood with this chip? Generally, most of the CTC isolation devices don’t require any pretreatment step.
  5. Page 6, Line 242. Please provide the essential fluorescent images of the captured cancer cells.
  6. Page 6, Line 243. Table S1 looks like that has no connection with this part. Please double-check Table S1 and this part.
  7. Page 7, Line 258. How are the GNPs digested by the MMP9? The whole GNPs are digested or just part of the GNPs? Can you provide more details and evidence about how the GNPs were digested? And what changes happen on the GNPs?
  8. Page 8 Line 276. Do you compare the cell viability and proliferative performance with cells without any processed? Please provide more data to support your conclusion.
  9. Figure 6A, why is the WBC is much larger than CTCs? It is believed that CTCs are larger than WBCs.
  10. There are some mistakes in the reference part. Please double-check.

Reviewer 3 Report

In this study, authors have reported multi-functional gelatin nanoparticles modified chip for enhanced capture and non-destructive release of circulating tumor cells. The concept of the manuscript is good, and the manuscript was written in a good manner but to their concept, additional results are required. However, authors should address the following comments before recommending them for publication.

  1. Figure 1. Schematic representation of the legend should be corrected or else explain the difference on it.
  2. Line no. 99. Please correct it “Propidium iodide”
  3. Line no.114, Please correct as “cultured” instead of “captured”. 
  4. Authors should provide the AFM image for nanoparticles coated chip to prove coating efficiency.
  5. How does this study confirm the streptavidin and biotinylated antibody conjugation?
  6. This manuscript contains so many merged words. Please correct it all appropriately.

Round 2

Reviewer 1 Report

The issues addressed before have been well corrected.

Reviewer 2 Report

Agree to publish on Micromachines

Reviewer 3 Report

-